# Biochar Reduces the Stability of Soil Aggregates during Intensive Leaching Experiment

**Nikola Teutscherova** [1],*, **Bohdan Lojka** [1] , **Marta Benito** [2], **Alberto Masaguer** [2] **and Eduardo Vázquez** [3],*

[1] Department of Crop Sciences and Agroforestry, Faculty of Tropical Agrisciences, Czech University of Life Sciences Prague, 16500 Prague, Czech Republic; lojka@ftz.czu.cz

[2] Departamento de Produccón Agraria, Escuela Técnica Superior de Ingeniería Agronómica, Alimentaria y de Biosistemas, 28040 Madrid, Spain; marta.benito@upm.es (M.B.); alberto.masaguer@upm.es (A.M.)

[3] Department of Soil Biogeochemistry and Soil Ecology, University of Bayreuth, 95448 Bayreuth, Germany

* Correspondence: teutscherova@ftz.czu.cz (N.T.); eduardo.vazquez@uni-bayreuth.de (E.V.)

**Abstract:** The interplay of different mechanisms shaping the biochar impact on soil structure remains relatively unexplored. We investigated the impact of biochar application to two contrasting soils on the stability of soil aggregates under an intensive intermittent leaching regime. A greenhouse experiment was established using PVC columns filled with 500 g soil from an Acrisol or Calcisol amended with three biochar applications (0, 1 and 2% *w/w*). The columns were watered weekly (100 mL) during two leaching cycles (each lasting 10 weeks). The amount of leached base cations, the stability of 1–2 mm aggregates fraction and soil chemical properties were determined. Biochar enhanced the leaching of the studied cations, but the content of base cations and effective cation exchange capacity remained higher in the biochar-amended Acrisol when compared to control soil. In both soils, biochar reduced the amount of water-stable aggregates, which seemed to be attributed to the increase of K in the exchange complex in the Acrisol while no significant correlation was detected between aggregation in Calcisol and other variables. The negative impact of biochar on soil aggregation is likely linked to higher sensitivity of biochar-amended soils to aggregate disruption under changing moisture conditions caused by frequent and intensive leaching events. These results highlight the gaps in our understanding of biochar impact on soil aggregation, which have implications for soil erodibility or restoration of degraded lands under changing climate.

**Keywords:** acrisol; aggregates; base cations; biochar; calcisol; leaching

## 1. Introduction

The capacity of biochar to reduce soil bulk density and increase soil porosity has been widely accepted in a wide range of soils [1]. Many authors have also reported enhanced formation of water-stable aggregates (WSA) after biochar application to soil [1]. The improvement of soil structure has been linked to increased microbial activity and aggregate stabilization by polysaccharides from microbial metabolism [2,3], and to the adsorption of organic matter onto biochar surface [4]. Thus, owing to its high carbon (C) content, biochar application to soil is not only a powerful way to sequester organic C, but could also be a promising strategy to improve soil structure, which is an essential step in preventing soil erosion and reducing nutrient losses.

Nevertheless, the effects of biochar on WSA formation described in the literature are rather inconsistent and seem to depend on soil types, pyrolysis conditions and feedstock materials for biochar production, as well as their interaction [5]. For instance, biochar had a lower effect on soil aggregation

in sandy or sandy loam soils when compared to silty or clay soils [5–8] because soils with finer texture and higher surface area likely provide more reaction sites for soil aggregation [5].

Inconsistencies have also been found between incubation studies, soil column experiments and field trials [9]. The majority of studies was performed under controlled laboratory conditions with optimal and constant moisture content for microbial growth and activity [2,7,10–13], which are generally reflected in enhanced soil aggregation. However, the results of such studies do not necessarily reflect field conditions, where soil moisture fluctuates, soil nutrients can be leached and the microbial activity can be limited by less favorable conditions. Therefore, it can be assumed that the experimental design plays a crucial role in determining the outcome of biochar application on soil WSA formation [8,9,14], which could lead to biased conclusions about the impact of biochar on soil aggregation.

According to the production temperature and feedstock material, biochar may differ considerably in the content of ashes and available nutrients [14], which can influence the composition of cation exchange complexes. The content of some cations, especially potassium (K), can be very high in fresh biochar. Although ash-derived nutrients are generally considered readily soluble and easily leached from the soil profile [15], the application of high amounts of monovalent cations can lead to a saturation of exchange complexes by these ions and result in displacement of divalent cations such as calcium (Ca) and magnesium (Mg) on the exchange sites. Both the losses of Ca and Mg (divalent ions) as well as the application of large amounts of K (monovalent) have been observed to be detrimental for soil aggregation [16,17]. Previous leaching experiments have revealed contrasting effects of biochar application on cation leaching [18–21] as well as on soil cation content and cation exchange capacity (CEC) after leaching events. Besides the clear importance of nutrient content and leaching from the agronomic point of view, changes in the composition of exchange complexes may also exert strong impacts on soil structure. Therefore, the understanding of the factors and mechanisms involved in cation leaching after biochar application and the evaluation of leaching impacts on soil aggregation seem to be crucial for accurate predictions of soil structure dynamics after biochar application.

The changes in soil aggregation after biochar application also depend on soil moisture regime (constant soil moisture vs. drying–rewetting cycle). Generally, drying–wetting cycles have been found to reduce soil aggregation because of aggregate slacking upon rewetting [22]. Grunwald et al. [23] demonstrated that biochar application under a drying–rewetting cycle decreased the stability of soil macroaggregates in comparison to the control treatment without biochar application, although the involved mechanisms were not completely addressed.

Despite the rapidly increasing number of studies focusing on the impacts of biochar on soil aggregation, recent literature reveals great inconsistencies in the results, probably due to the high number of simultaneous biochar-induced changes in soil and the variability of biochar types, soils and experimental set-ups. Therefore, understanding biochar effects on WSA under less optimal conditions for soil aggregation but more similar to field conditions is necessary to disentangle the involved mechanisms and the potential effects under field conditions. The aim of the present study was to evaluate how biochar application to two contrasting soils affects the content of WSA in a column leaching experiment [24], where high watering rates likely ensured base cation leaching and intermittent reducing conditions (caused by drying–rewetting events). Such conditions in turn likely influenced the content of WSA in soil either directly by modification of soil chemical and physical properties or indirectly via modification of microbial activity. We hypothesized that aggregate disruption after biochar application under a leaching regime will be higher in an acid Acrisol due to the coarse texture of the soil and its initially low CEC and Ca and Mg content. In a Calcisol, with high CEC, divalent nutrient content and soil pH, as well as clayey loamy texture, we hypothesize no or a low effect of biochar on soil aggregation.

## 2. Materials and Methods

### 2.1. Soil and Biochar Characterization

The properties of both contrasting soils used in the present study can be found in [24] and in Table 1. Briefly, acid Acrisol was collected from Cañamero's raña formation in SW Spain. This soil is characterized by sandy loam texture in the topsoil, low base saturation and low pH. The second soil selected was classified as Calcisol and obtained from "La Chimenea" Field Station near Aranjuez (Madrid, Spain). It has a high pH and carbonate content and a clayey loamy texture (Table 1). The samples were collected from the topsoil layer (20 cm), transported to the laboratory and sieved (<5 mm) at field-moist conditions. Biochar was produced from holm oak pruning residues in a batch system at a temperature of 600 °C under a restricted oxygen environment and then crushed and sieved (<2 mm) [24].

**Table 1.** Selected soil and biochar properties.

| Soil Properties | Acrisol | Calcisol | Biochar Properties | |
|---|---|---|---|---|
| pH | 5.65 | 8.00 | pH | 10.2 |
| Electric conductivity ($\mu$S cm$^{-1}$) | 49.7 | 570 | Electric conductivity ($\mu$S cm$^{-1}$) | 940 |
| CEC (cmol$_c$ kg$^{-1}$) | 2.73 | 8.84 | TC (%) | 68.2 |
| TOC (g kg$^{-1}$) | 25.8 | 9.55 | TN (%) | 0.67 |
| Carbonate content (%CaCO$_3$) | n.p. | 21.9 | C$_{ox}$ (%) | 4.70 |
| TN (g kg$^{-1}$) | 1.28 | 0.90 | Ash content (%) | 3.49 |
| WSC (mg kg$^{-1}$) | 78.3 | 29.1 | Carbonates content (%CaCO$_3$) | 11.9 |
| WSN (mg kg$^{-1}$) | 19.0 | 49.2 | WSC (mg kg$^{-1}$) | 149 |
| Field moisture capacity (%) | 16.9 | 18.3 | WSN (mg kg$^{-1}$) | 93.4 |
| Sand (%) | 80.1 | 29.0 | CEC (cmol$_c$ kg$^{-1}$) | 35.1 |
| Silt (%) | 6.10 | 42.0 | NH$_4$-N sorption (mg NH$_4$-N g$^{-1}$) | 2.22 |
| Clay (%) | 13.8 | 29.0 | NO$_3$-N sorption (mg NO$_3$-N g$^{-1}$) | n.s. |

CEC, cation exchange capacity; TOC, total organic carbon; TN, total nitrogen; TC, total carbon; C$_{ox}$, dichromate oxidizable organic C; WSC, water soluble carbon; WSN, water soluble nitrogen; n.p., not present, n.d., not detectable.

### 2.2. Experimental Design and Column Preparation

Columns were prepared as described in a previous study [24]. Briefly, both soils (Acrisol and Calcisol) were amended with 1% (B1) and 2% (B2) of biochar and soils without biochar addition were used as controls (B0). Four replications of each soil and biochar mixture were packed in nontransparent PVC columns (5 cm diameter and 30 cm height) to a bulk density of approximately 1.3 g cm$^{-3}$, which corresponds to the bulk density of the studied soils. Bulk density was adjusted in the control soils and the same pressure was applied to compact the biochar-treated soils. All columns were fitted with fiber mesh and a funnel on the bottom. A 5 cm layer of gravel and acid-washed sand was placed inside each column to prevent soil losses. Control columns (without biochar amendment) received 500 g of soil and columns amended with biochar were filled equivalently with 500 g soil and the corresponding amount of biochar (a total weight of 505 and 510 g for B1 and B2, respectively). Considering the soil bulk density (1.3 g cm$^{-3}$) and a soil layer of 20 cm (the height of soil in the columns), the amount of biochar added was equivalent to 26 and 52 Mg ha$^{-1}$ in B1 and B2, respectively.

### 2.3. Leaching Experiment

Soil columns were wetted to 40% of water holding capacity (WHC), and preincubated for one week. After seven days, moisture content was increased to 60% WHC. The PVC columns were placed in a randomized design in a custom-made wooden rack located in a controlled greenhouse (temperature adjusted to 25 °C). No leaching was observed during the preparatory phase.

Columns were leached weekly with 100 mL of deionized water for ten consecutive weeks (leaching events) and the leachate was collected during 24 h after each leaching event. The amount of

water applied was comparable to the watering regime applied in other studies [18,25], and considering the column soil surface, each leaching event was equivalent to 51 mm of rainfall. The large amount of applied water simulated the winter conditions in the study area with groundwater level at soil surface for several months. After ten weeks, columns were left to dry for four weeks, followed by rewetting of the columns to 60% of their WHC. One week after, columns were subjected to the first leaching event of the second leaching cycle. At the end of the second leaching cycle (ten weeks), a subsample of the soil from each column was air-dried for the analysis of soil properties. During the lifespan of the experiment (24 weeks), 2000 mL of water were added to each soil column, which was the equivalent of 1.3 and 2.7 years of rainfall in the area where Acrisol and Calcisol were collected, respectively.

*2.4. Analytical Methods*

At the end of the experiment, soil pH and electric conductivity were determined in a soil:water suspension (1:2.5, *w/v*). The content of soil organic matter (SOM) was determined by loss on ignition at 540 °C. The content of exchangeable base cations (Ca, Mg, K, Na) was quantified by atomic absorption spectroscopy (AAnalyst 400, PerkinElmer, Wellesley, MA, USA) in Mehlich III extraction solution [26]. Soil-available P was extracted using the Bray-1 method in the Acrisol and using the Olsen method in the Calcisol, and analyzed colorimetrically [27]. After the experiment, the soil was air-dried and gently sieved through 2 mm and 1 mm sieves and the weights of all fractions were noted. An 4 g aliquot of the 1–2 mm aggregate fraction was wetted by capillarity for 15 min on cotton pads to prevent aggregate rupture by slacking upon rewetting. The content of water-stable aggregates (WSA) was then determined by wet sieving through a 250 µm sieve [28]. The sand and biochar particle contents were accounted for by dispersing the aggregates with sodium hexametaphosphate (5 g $L^{-1}$). The WSA was calculated as the weight of stable aggregates divided by the sum of stable and unstable aggregates and subtracting the sand and biochar content. The aggregates were expressed as a percentage of WSA, as well as in grams of WSA per kg of soil. Soil acidity and Aluminum (Al) content (only Acrisol) were determined using the method of [29] by adding NaOH and KF solution to the soil extracts and by titration of excess $OH^-$ with $H_2SO_4$. The effective cation exchange capacity (ECEC) of the soil at the end of the experiment was calculated as the sum of exchangeable $H^+$ and $Al^{3+}$ and exchangeable base cations [30]. The concentration of cations in the leachate was measured with atomic absorption spectroscopy. The pH and electric conductivity (EC) of the leachate were determined by a pH- and conductimeter, respectively.

*2.5. Statistical Analyses*

The effects of biochar application on cumulative leaching losses and final soil properties were analyzed by a one-way ANOVA separately for each soil. A post hoc least significant difference (LSD) test was used to evaluate the differences between the three biochar application rates ($p < 0.05$). Data were checked for normality and homogeneity of variance prior to analyses and transformed (log + 1) when necessary prior ANOVA. Bivariate correlations between $WSA_{1-2\,mm}$ expressed as g $kg^{-1}$ and final soil properties were determined for each soil by the Pearson's correlation coefficients. All the analyses were performed using SPSS 22 (IBM SPSS, Inc., Chicago, IL, USA) software.

## 3. Results

*3.1. Cumulative Leaching Losses of Base Cations*

The leaching of base cations (Ca, Mg, K) was higher in the Calcisol during both leaching cycles (Figure 1 and Figure S1) and increased by both biochar application rates in both soils. The leaching of Na was unaffected by biochar addition in Acrisol; meanwhile in Calcisol, Na concentration in the leachate was significantly higher in B2 than in B0 during the first leaching cycle, while in the second cycle, the Na losses were below the detection limit (Figure 1 and Figure S1).

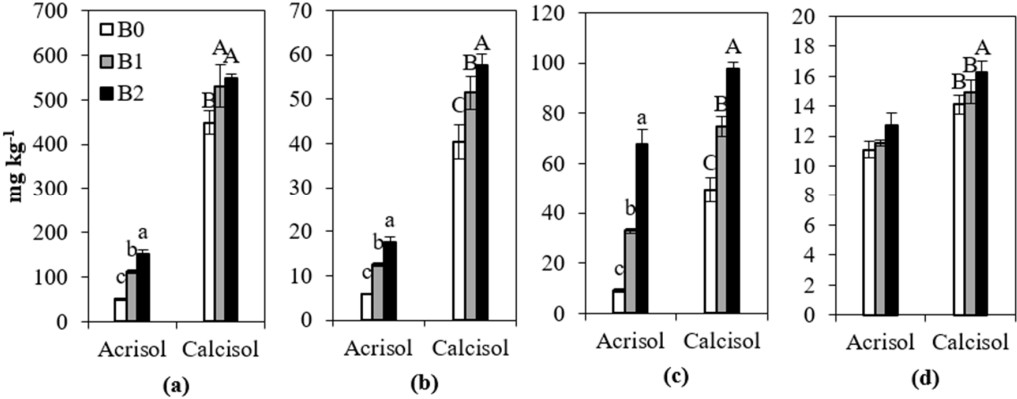

**Figure 1.** Cumulative leaching losses of Calcium (**a**), Magnesium (**b**), Potassium (**c**) and sodium (**d**) from Acrisol and Calcisol. B0, control without biochar addition; B1 and B2, 1% and 2% biochar addition, respectively. Bars indicate standard error of the mean (n = 4). Different letters indicate significant differences at $p < 0.05$.

### 3.2. Final Soil Properties

At the end of the leaching experiment, the pH of the Acrisol without biochar amendment dropped by almost 1.8 units during the experiment, while the pH of the Calcisol increased by 0.1 pH unit (Table 2). The pH in the biochar-amended Acrisol was significantly higher than in the control soil at the end of the experiment. Nevertheless, the pH of the Calcisol was comparable between biochar treatments and was only slightly higher in B2 when compared to B0 ($p < 0.05$) (Table 2). At the end of the experiment, the addition of both biochar application rates increased the SOM content in both soils (Table 2). The effect on SOM was relatively higher in Calcisol, where B2 was 118% higher than B0, when compared to Acrisol, where B2 was 45% higher than B0.

**Table 2.** The soil properties at the end of the leaching experiment. Mean ± standard error (n = 4).

| | pH | SOM % | P mg kg$^{-1}$ | Ca | Mg | K cmol$_c$ kg$^{-1}$ | Al | ECEC | K/ECEC % |
|---|---|---|---|---|---|---|---|---|---|
| *Acrisol* | | | | | | | | | |
| B0 | 3.88 ± 0.02 [c] | 5.08 ± 0.09 [c] | 29.3 ± 0.7 [c] | 0.96 ± 0.02 [c] | 0.11 ± 0.01 [c] | 0.05 ± 0.00 [c] | 1.11 ± 0.04 [a] | 2.71 ± 0.10 [c] | 1.98 ± 0.12 [b] |
| B1 | 4.51 ± 0.01 [b] | 6.12 ± 0.20 [b] | 38.5 ± 1.4 [b] | 2.56 ± 0.04 [b] | 0.18 ± 0.01 [b] | 0.11 ± 0.01 [b] | 0.01 ± 0.00 [b] | 3.42 ± 0.15 [b] | 3.34 ± 0.21 [a] |
| B2 | 5.06 ± 0.04 [a] | 7.38 ± 0.08 [a] | 50.6 ± 1.0 [a] | 3.82 ± 0.07 [a] | 0.22 ± 0.01 [a] | 0.14 ± 0.01 [a] | 0.08 ± 0.04 [b] | 4.45 ± 0.08 [a] | 3.20 ± 0.13 [a] |
| *Calcisol* | | | | | | | | | |
| B0 | 8.17 ± 0.01 [A] | 1.83 ± 0.01 [C] | 80.9 ± 1.9 [A] | 24.44 ± 0.15 | 1.34 ± 0.01 [A] | 0.49 ± 0.01 [B] | - | 26.3 ± 0.2 | 1.86 ± 0.03 [C] |
| B1 | 8.19 ± 0.01 [A] | 2.90 ± 0.04 [B] | 63.4 ± 2.5 [B] | 24.55 ± 0.37 | 1.27 ± 0.02 [B] | 0.57 ± 0.00 [A] | - | 26.4 ± 0.4 | 2.16 ± 0.04 [B] |
| B2 | 8.14 ± 0.01 [B] | 4.00 ± 0.04 [A] | 77.8 ± 3.6 [A] | 24.12 ± 0.14 | 1.29 ± 0.01 [B] | 0.62 ± 0.00 [A] | - | 26.0 ± 0.1 | 2.38 ± 0.08 [A] |

SOM, soil organic matter; P, phosphorus; Ca, calcium; Mg, magnesium; K, potassium; Al, aluminum; ECEC, effective cation exchange capacity. B0, control without biochar addition; B1 and B2, 1% and 2% biochar addition, respectively. Different lowercase and uppercase letter indicate statistically significant differences ($p < 0.05$) among biochar doses in Acrisol and Calcisol, respectively.

The content of Ca, Mg and K remained higher in the Calcisol when compared to the Acrisol (Table 2). In Acrisol, the application of both doses of biochar significantly increased the final content of Ca, Mg and K (Table 2). In Calcisol, the Ca content was unaffected by biochar addition, while Mg and K contents were reduced and increased, respectively (Table 2). The Na content in the soil at the end of the leaching experiment was below the detection limit in both soils.

Biochar reduced the Al content in the Acrisol and soil acidity (Table 2). The ECEC was significantly higher in the Calcisol than in the Acrisol, and was increased by biochar application in the Acrisol (Table 2). The relation between K and ECEC (K/ECEC) was increased by biochar application in both soils (Table 2).

The effect of biochar application on available P varied between the soils used. In the Acrisol, P content was increased by both biochar applications, while in the Calcisol, P content was lowered by B1 and unaffected by B2, when compared to B0 (Table 2).

### 3.3. Water-Stable Aggregates

The stability of soil aggregates was higher in the Acrisol than in the Calcisol. In Calcisol, the content of $WSA_{1-2\,mm}$ (expressed as % or g kg$^{-1}$ soil) was reduced by the highest biochar application rate (Table 3). In Acrisol, only the % of $WSA_{1-2\,mm}$ was significantly lower in B2 than in B0, and $WSA_{1-2\,mm}$ expressed as g kg$^{-1}$ soil was reduced by biochar addition, although the differences were not significant (Table 3). The $WSA_{1-2\,mm}$ expressed as g kg$^{-1}$ was related to K/CEC in the Acrisol (R = −0.663; $p$ = 0.019), but it could not be explained by any of the tested variables in Calcisol.

**Table 3.** Percentage of water stable aggregates ($WSA_{1-2\,mm}$) and content of WSA per kg soil. Mean ± standard error (n = 4).

| | $WSA_{1-2\,mm}$ % | $WSA_{1-2\,mm}$ g kg$^{-1}$ soil |
|---|---|---|
| *Acrisol* | | |
| B0 | 91.3 ± 2.7 [a] | 164 ± 11 [a] |
| B1 | 84.9 ± 3.0 [ab] | 139 ± 15 [a] |
| B2 | 81.4 ± 4.5 [b] | 143 ± 10 [a] |
| *Calcisol* | | |
| B0 | 8.96 ± 1.24 [A] | 49.5 ± 3.9 [A] |
| B1 | 9.84 ± 0.54 [A] | 52.7 ± 5.8 [A] |
| B2 | 5.62 ± 1.33 [B] | 30.0 ± 5.9 [B] |

B0, control without biochar addition; B1 and B2, 1% and 2% biochar addition, respectively. Different lowercase and uppercase letter indicate statistically significant differences ($p$ < 0.05) among biochar doses in Acrisol and Calcisol, respectively.

## 4. Discussion

Biochar application reduced the content of WSA in both studied soils with contrasting properties in an intensive leaching column experiment. The reduced aggregation was observed as a reduction of percentage of $WSA_{1-2\,mm}$ in both soils, as well as in grams of $WSA_{1-2\,mm}$ kg$^{-1}$ soil in Calcisol. Thus, the observed results could not be explained by a "dilution effect" caused by biochar application, as the "dilution effect" could affect the $WSA_{1-2\,mm}$ when expressed as g kg$^{-1}$ soil, but not the % of $WSA_{1-2\,mm}$. The discrepancy between our results and [3], where WSA were increased in both soil types after the application of the same biochar in an incubation study, could be caused by (i) intensive leaching of nutrients and/or changes in the proportion of monovalent and divalent ions; (ii) changes in microbial substrates directly hindering microbial activity in soils; and (iii) differences in soil aggregate resistance to frequent fluctuations of soil moisture. Furthermore, the content of soil aggregates and the amount of WSA per kg of soil decreased in all treatments from the initial WSA contents (data not shown), which indicates that biochar application promoted higher aggregate disruption rather than reduced WSA formation during the leaching experiment in comparison to laboratory incubation.

### 4.1. Base Cations Dynamics and Exchange Complex

Biochar has been suggested by many authors to prevent nutrient losses due to adsorption of soil nutrients onto the biochar surface [21,22,27]. However, depending on the feedstock material and production temperature, biochar itself can contain a variable quantity of cations in the ash fractions, which are readily soluble [15]. The application of biochar caused higher leaching of Ca, Mg, K and Na when compared to nonamended soils, similarly to the results obtained by [18]. The capacity of biochar to supply nutrients and/or affect nutrient stoichiometry in biochar-amended soils under leaching requires more attention in future studies.

Despite the higher Ca, Mg and K leaching, higher contents of available Ca, Mg and K and higher soil pH were found in the biochar-amended Acrisol when compared to the control at the end of the experiment. These results correspond with other studies [21], where higher Ca and K contents were observed in the biochar-amended soil after leaching in a loamy sand acid soil, similar to the Acrisol

used in the present study. However, the influence of the biochar application on the final cation content in the Calcisol was more erratic, with no effect on Ca, a trend of lower Mg content and higher K content with biochar application. These differences between both soils in the effect of biochar on the final cation content could be linked to the contrasting ECEC of both soils: while in the Acrisol the lower initial ECEC increased with increasing biochar application rate, biochar application had no effect on the ECEC in the Calcisol (whose ECEC was several times higher than that of the Acrisol). Many studies observed an increase of soil ECEC and base cation content after biochar application, mainly to acidic soils in laboratory experiments where no nutrients could be leached [31–33]. Nevertheless, the effect of biochar on CEC and Ca content in alkaline soils is more erratic in laboratory experiments [34] or under field conditions where leaching of nutrients was likely to occur; [35] detected no impact of biochar on available base cations or ECEC in alkaline soil, similar to our column leaching study.

### 4.2. The Implications of Intensive Leaching for Soil Aggregation

The formation of soil aggregates is a result of flocculation and cementation processes. While flocculation is affected mainly by pH, EC and Na content, cementation depends on the amount and quality of binding agents, such as Ca carbonate, Mg carbonate, gypsum, sesquioxides, clay particles and SOM [36]. Both flocculation and cementation were likely affected in the present study, where soil properties were modified by biochar application and by intensive leaching.

The bridging effect of Ca between clay and SOM is of particularly high importance in soil aggregate formation. Therefore, biochar containing high amounts of Ca may be more efficient in soil structure rebuilding, especially when compared to biochar with higher quantities of monovalent ions. However, biochar often contains a large amount of K, which has been known to cause aggregate disruption [16] due to its low charge-to-size ion ratio. The leaching of divalent ions and the alteration of the exchange complex composition (as observed by the increase of K/ECEC in both soils with biochar application) could contribute to the differences between (increased) aggregate stability in the incubation study [3] and the present leaching experiment. This assumption is also confirmed by the correlation analysis where a negative relationship between K/ECEC and $WSA_{1-2\,mm}$ expressed as g kg$^{-1}$ in the Acrisol was detected. However, no significant correlations were observed between WSA in Calcisol and the measured soil properties. In summary, the addition of biochar under leaching conditions led to an alteration of the cation contents and ratios that could explain the decrease of soil aggregation in Acrisol.

In the present study, immediately after watering the soil columns, soils became saturated with water at least for a short period of time. After drainage of excessive water (within 24 h), soil columns were left without any irrigation until the next leaching event (six days). Although the soil moisture was not measured prior to water application, fluctuations in the soil water content must have occurred. Such conditions of continuous drying/rewetting could lead to the disruption of soil aggregates [22] and more intensive microbial decay of SOM (compared to continuous watering regimes) [37], which could be further stimulated in the biochar-amended Acrisol due to soil pH neutralization [3,13]. Many authors have related higher microbial activity with enhanced formation of WSA [2] unless the microbial substrates and products are lost from the soil profile by leaching, as likely occurred in the present experiment where high content of water soluble C and N should translate into higher microbial activity.

In the same experiment as the present study [24], increased N losses from the Acrisol were observed after biochar application, which was attributed to enhanced initial N mineralization resulting from the amelioration of the soil chemical properties (acidity and Al toxicity), causing stimulation of labile SOM pools under drying–rewetting cycles [37] or to the flush of N released after the initial rupture of soil aggregates, particularly in the biochar treatments.

Soil pH neutralization is also linked with changes in microbial communities and shifts towards bacteria-driven processes [38]. Soil bacteria actively decompose SOM and build soil microaggregates [39], which are then glued together to form larger aggregates by the action of plant roots and soil fungi. Furthermore, fungal communities and higher SOM contents have been found to increase the resistance of aggregates to slaking and to reduce the nutrient leaching [23,40]. Thus, the pH

neutralization of the Acrisol by biochar could lead to a shift towards a lower fungi-to-bacteria ratio [41] as observed in the Acrisol under study [42], where a lower fungi-to-bacteria ratio and reduced abundance of fungi were observed. Such alterations may in turn lead to reduced formation of WSA and their resistance to slacking, in contrast to B0 [22,42]. Clearly, the simultaneous evaluation of biochar-driven changes in physical, chemical and biological properties requires more attention in future studies.

The depletion of native SOM in the biochar treatments by stimulated mineralization linked with leaching of nutrients and organic matter could reduce the number of binding agents capable to rebuild the aggregates broken by slacking. The combination of these processes is common under field conditions, where drought periods alternate with rainfall events and, eventually, with leaching. However, most of the studies aiming to evaluate the impact of biochar aggregation have been performed under optimal and constant moisture content without leaching, which could explain the inconsistency of our results or other field studies with other incubation or pot studies. Therefore, studies under field conditions, as well as under variable or suboptimal moisture conditions, are required to disentangle the changes induced by biochar application to soil and their impact on soil structure and nutrient leaching. Furthermore, because soil macroaggregates are stabilized mainly by temporary organic matter, including fungal hyphae and plant roots [43], the results of the present study may not reflect field conditions where the role of plant roots and associated rhizosphere microbiota is crucial.

## 5. Conclusions

The impact of biochar application to two contrasting soils subjected to frequent and intensive leaching was evaluated in a column leaching study for 25 weeks. Biochar increased base cations contents in the leachate, which indicates that cations contained in the biochar exceeded the sorption capacity of biochar. We observed a reduced content of WSA at the end of the experiment in both soils (acid carbon-rich Acrisol and alkaline Calcisol poor in organic matter), with a lower WSA content in biochar-containing treatments. The reduced WSA content after biochar application was likely caused by intensive leaching of divalent base cations and linked changes in cation exchange complexes. Although the mechanisms involved in the decrease of WSA by biochar application have not been fully addressed in the present study, our results encourage the evaluation of biochar effects on soil aggregation and soil nutrient contents under field condition with leaching and soil moisture fluctuations.

**Supplementary Materials:** The following are available online at http://www.mdpi.com/2073-4395/10/12/1910/s1, Figure S1: Cumulative leaching losses of Ca, Mg, K and Na from Acrisol and Calcisol. Bars indicate standard error of the mean (n = 4).

**Author Contributions:** Conceptualization, N.T. and E.V.; methodology, N.T. and E.V.; investigation, N.T. and E.V.; resources, N.T., B.L., M.B., A.M. and E.V.; data curation, N.T. and E.V.; writing—original draft preparation, N.T. and E.V.; writing—review and editing, N.T., B.L., M.B., A.M. and E.V.; supervision, N.T. and E.V.; funding acquisition, A.M. All authors have read and agreed to the published version of the manuscript.

**Funding:** This research was funded by AGRISOST-CM (S2013/ABI-2717) from the Comunidad de Madrid and cofunded by the European Structural and Investment Funds. Nikola Teutscherova thanks Cátedra Rafael Dal-Re/TRAGSA for their financial support and Eduardo Vázquez thanks the Spanish Ministry of Education for his FPU fellowship. Financial support was also obtained from Integral Grant Agency of the Czech University of Life Science Prague (no. 20205003).

**Acknowledgments:** Special thanks also belong to Roman Zurita for his assistance and practical support during the establishment of the experiment.

**Conflicts of Interest:** The authors declare no conflict of interest. The funders had no role in the design of the study; in the collection, analyses, or interpretation of data; in the writing of the manuscript, or in the decision to publish the results.

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
