# Peer review of "Biochar Reduces the Stability of Soil Aggregates during Intensive Leaching Experiment"

_agronomy, doi:10.3390/agronomy10121910_

Round 1
Reviewer 1 Report
This paper has analyzed the effects of biochar on soil aggregation in the intensive leaching scenarios. The authors have conducted a good work with well-designed experiments and analysis. I only have a few questions to discuss with the authors.
- Lines 134-136, the authors have mentioned that the added water during the 24-week experiment equaled to 1.3 and 2.7 years of the rainfalls in the soil sampling areas. In my opinion, the leaching scenario created in this study was too intensive about 2-4 times of the field rainfalls. Could the authors provide more information on the related weathering field conditions reported in other literatures to support the aim of this study?
- In the discussion section, the effects of SOMs and microbial communities on soil aggregation were discussed. However, in the results section, the SOM contents and the microorganism contents of the soil samples were not determined. It makes the discussion and the conclusion less reliable. At least, the soil SOM contents could be easily analyzed and supplied.
Author Response
Reviewer 1
This paper has analyzed the effects of biochar on soil aggregation in the intensive leaching scenarios. The authors have conducted a good work with well-designed experiments and analysis. I only have a few questions to discuss with the authors.
- Lines 134-136, the authors have mentioned that the added water during the 24-week experiment equaled to 1.3 and 2.7 years of the rainfalls in the soil sampling areas. In my opinion, the leaching scenario created in this study was too intensive about 2-4 times of the field rainfalls. Could the authors provide more information on the related weathering field conditions reported in other literatures to support the aim of this study?
- The groundwater level in the study area is usually at soil surface from autumn to spring due to high rainfall and inadequate soil drainage. Therefore, while the amount of applied water may seem to be excessive considering rainfall distribution during the whole year, it well described the winter conditions. Similar amounts of water were applied in other studies [1,2,3]
Kuo, Y. L., Lee, C. H., & Jien, S. H. (2020). Reduction of nutrient leaching potential in coarse-textured soil by using biochar. Water, 12(7), 2012.
Laird, D.; Fleming, P.; Wang, B.; Horton, R.; Karlen, D. Biochar impact on nutrient leaching from a Midwestern agricultural soil. Geoderma 2010, 158, 436–442, doi:10.1016/j.geoderma.2010.05.012.
Teutscherova, N.; Houška, J.; Navas, M.; Masaguer, A.; Benito, M.; Vazquez, E. Leaching of ammonium and nitrate from Acrisol and Calcisol amended with holm oak biochar: A column study. Geoderma 2018, 323, 136–145, doi:10.1016/j.geoderma.2018.03.004.
In the discussion section, the effects of SOMs and microbial communities on soil aggregation were discussed. However, in the results section, the SOM contents and the microorganism contents of the soil samples were not determined. It makes the discussion and the conclusion less reliable. At least, the soil SOM contents could be easily analyzed and supplied.
- The content of SOM was added to the manuscript and discussion was edited to be less speculative and more supported by the experimental data.
Reviewer 2 Report
This is interesting research. The sections of the introduction and materials/methods are well described.
However, there is a missing point as follow:
It’s understandable that they focused on the chemical and physical properties. However, the authors mentioned the impact of biochar addition on soil microbial activity. In this paper, they didn’t relate this with their own results. And it gives a bit the impression of the speculation. They should improve this part by using their own data. For instance, in Table 1, the authors represent WSC and WSN which can be nutrients for microbes.
More detailed comments are shown below a
Line 40-41: “soil types, pyrolysis conditions and feedback materials for biochar production as well as their interaction.
Line 280 and Line 35: The authors mentioned twice the importance of polysaccharide by citing other works. What about their own data? The authors showed the WSC and WSN
Consistency of using nouns with “plural” or “single” is important (soil type, condition, material).
Line 109-110: Why the authors chose 1% and 2% of biochar addition?
Table 2. Some of the properties are not represented with units.
Table 3. Missing letters for statistical analysis in WSA kg-1 soil.
Author Response
Reviewer 2
This is interesting research. The sections of the introduction and materials/methods are well described.
However, there is a missing point as follow:
- It’s understandable that they focused on the chemical and physical properties. However, the authors mentioned the impact of biochar addition on soil microbial activity. In this paper, they didn’t relate this with their own results. And it gives a bit the impression of the speculation. They should improve this part by using their own data. For instance, in Table 1, the authors represent WSC and WSN which can be nutrients for microbes.
We would like to thank the reviewer for his/her suggestions. We reduced the somewhat speculative parts from the discussion. While we do agree that WSC and WSN data from table 1 could be useful in explaining increased higher microbial activity, which would translate into high WSA, we found the opposite in the present study. As unfortunately the WSC and WSN (or directly the microbial biomass) were not determined at the end of the experiment, no conclusions could be made. We, hopefully, made the discussion less speculative now.
More detailed comments are shown below a
- Line 40-41: “soil types, pyrolysis conditions and feedback materials for biochar production as well as their interaction.
- Corrected
- Line 280 and Line 35: The authors mentioned twice the importance of polysaccharide by citing other works. What about their own data? The authors showed the WSC and WSN
We slightly modified the parts the reviewer is referring to. Unfortunatelly, we do not have WSC and WSN data from the end of the experiment or from the leachate. While WSC and WSN values of biochar (Table 1) could indicate higher microbial activity, it cannot be ruled out from our dataset. Hence, the reference to polysaccharates and microbial activity is more a possible explanation of the found results, as they could not be satisfactorily explained by measured variables. We hopefully made it better understandable from the text.
- Consistency of using nouns with “plural” or “single” is important (soil type, condition, material).
- The text was checked for inconsistencies
- Line 109-110: Why the authors chose 1% and 2% of biochar addition?
- The application rates 1% and 2% (26 and 52 t ha-1) were selected based on the literature review as well as on the recommendations of the application rate around 50 t ha-1 for C trading and GHG emissions mitigations in most agricultural systems [1]. [2] indicated that at 50 t ha-1 application rate, the mean crop productivity increase would be around 28%. While the mean crop productivity increase was then reduced to 18% [3], the potential of application rate of 50 t ha-1 should be confirmed. The lower application rate was selected based on the indications that application rates higher than 5 t ha-1 can already increase crop productivity [3], hence, their impact on soil processes should be understood. Furthermore, application rates 1 and 2% are widely used by other authors, which facilitates comparisons with other studies performed under different conditions.
Woolf, D.; Amonette, J.E.; Street-Perrott, F.A.; Lehmann, J.; Joseph, S. Sustainable biochar to mitigate global climate change. Nat. Commun. 2010, doi:10.1038/ncomms1053.
Jeffery, S.; Verheijen, F.G.A.; van der Velde, M.; Bastos, A.C. A quantitative review of the effects of biochar application to soils on crop productivity using meta-analysis. Agric. Ecosyst. Environ. 2011, 144, 175–187, doi:10.1016/j.agee.2011.08.015.
Lehmann, J.; Stephen, J. Biochar for Environmental Management: Science, Technology and Implementation; 2015; ISBN 9781844076581.
Lehmann, J.; Joseph, S.; Earthscan from Routledge. Biochar for environmental management : science, technology and implementation; Lehmann, J., Joseph, S., Eds.; second edition.; 2015; ISBN 9781134489534.
- Table 2. Some of the properties are not represented with units.
- SOM was added to the table 2 (as requested by other reviewer) and all variables have units now
- Table 3. Missing letters for statistical analysis in WSA kg-1 soil.
- Letters were added